# Balloon Pulmonary Angioplasty for Inoperable Chronic Thromboembolic Pulmonary Hypertension: Insights from a Pilot Low-Volume Centre Study and a Comparative Analysis with Other Centres

**DOI:** 10.3390/medicina60030461

**Published:** 2024-03-11

**Authors:** Taida Ivanauskiene, Sigitas Cesna, Egle Grigoniene, Lina Gumbiene, Aurelija Daubaraite, Kaste Ivanauskaite, Sigita Glaveckaite

**Affiliations:** 1Institute of Clinical Medicine, Clinic of Cardiovascular Diseases, Faculty of Medicine, Vilnius University, 01513 Vilniaus, Lithuania; sigitas.cesna@santa.lt (S.C.); egle.grigoniene@santa.lt (E.G.); lina.gumbiene@santa.lt (L.G.); sigita.glaveckaite@santa.lt (S.G.); 2Pulmonary Hypertension Referral Centre, Vilnius University Hospital Santaros Klinikos, 08661 Vilnius, Lithuania; 3Department of Psychology, University of Amsterdam (UvA), 1012 WP Amsterdam, The Netherlands; kaste.ivanauskaite@gmail.com

**Keywords:** balloon pulmonary angioplasty, inoperable chronic thromboembolic pulmonary hypertension, mean pulmonary artery pressure, pulmonary vascular resistance

## Abstract

*Background and Objectives*: The aim of this study was to evaluate the effectiveness and safety of balloon pulmonary angioplasty (BPA) in patients with inoperable chronic thromboembolic pulmonary hypertension (CTEPH) in the Vilnius Pulmonary Hypertension (PH) Referral Centre and to provide a comparative analysis with other centres. *Materials and Methods*: This study included all BPA procedures performed between 2019 and 2024 in a single tertiary centre. Invasive haemodynamic parameters and clinical variables were assessed at baseline; at the end of invasive treatment; and at the conclusion of follow-up, an average of 8.6 months after the last BPA. A literature review was also performed. *Results*: Twenty-six patients with inoperable CTEPH were enrolled. The mean age of the patients was 61.6 (40–80) years. Each patient underwent a mean of 3.84 (1–9) procedures. Follow-up data were available for 12 patients with an average of 6.08 (3–9) procedures. Mean pulmonary arterial pressure decreased by 32% (*p* < 0.001) and pulmonary vascular resistance by 41% (*p* = 0.001) at follow-up compared with the baseline measurements. There was also a significant 80% (*p* < 0.001) reduction in brain natriuretic peptide levels and a 30% (*p* = 0.04) increase in 6-min walk distance. The BPA procedures were generally safe in this low-volume centre setting, with only 17% of procedures having non-severe and non-fatal procedure-related complications. The most common complications included vessel dissection (10%), pulmonary vascular injury with haemoptysis (3%), and hyperperfusion pulmonary oedema (1%), which was successfully treated in all patients. *Conclusions*: The results of the present study demonstrate that the BPA procedure is an effective and safe treatment for individuals with inoperable CTEPH, being associated with significant improvements in hemodynamic parameters and functional capacity and a low risk of major complications in the low-volume tertiary PH centre setting.

## 1. Introduction

Chronic thromboembolic pulmonary disease (CTEPD) with or without pulmonary hypertension (PH) is a disease caused by persistent obstruction of the pulmonary arteries by organised fibrotic thrombi and associated microvasculopathy. Chronic thromboembolic pulmonary hypertension (CTEPH) is usually considered a complication of acute pulmonary embolism (PE), and it has been estimated that 0.56–3% of patients with acute PE develop CTEPH [1,2]. Patients with CTEPH are symptomatic with mismatched perfusion defects on pulmonary ventilation/perfusion (V/Q) scans and evidence of chronic, organised, fibrotic clots on computed tomography pulmonary angiography (CTPA) or digital subtraction angiography (DSA) after at least 3 months of therapeutic anticoagulation [3]. The incidence of CTEPH is 3.1–6.0 per million, and the prevalence is up to 25.8–38.4 per million [4]. The number of people diagnosed with CTEPH is increasing due to a better understanding of the disease and more active screening [3]. 

Pulmonary endarterectomy (PEA) is the first-line treatment for patients with CTEPH and, when performed in experienced centres, is associated with an in-hospital mortality of <5%, as well as haemodynamic, functional, and survival improvement [5,6,7,8]. However, surgery cannot be performed in certain situations, such as in the distal location of the disease (technically not feasible) and when the patient has comorbidities (poor risk/benefit ratio) [9]. As a result, approximately 40% of CTEPH patients are inoperable [8]. In addition, up to 25% of operated patients have residual or recurrent PH [10]. Such patients can be treated with PH-specific medical therapy, but the current evidence of haemodynamic and symptomatic improvement lacks long-term data [11,12]. 

Consequently, balloon pulmonary angioplasty (BPA) has recently emerged as a treatment option and is now recommended by the European Society of Cardiology (ESC) and the European Respiratory Society (ERS) for patients who are technically inoperable or have residual PH after PE and distal obstructions amenable to BPA (class IB recommendation) [3]. BPA is an interventional procedure that uses a specifically sized balloon to dilate organised thrombi and restore blood flow to the distal pulmonary arteries [13]. Evidence shows that BPA improves pulmonary haemodynamics, 6-min walk distance (6MWD), functional class, and quality of life by improving pulmonary circulation [9,14,15,16]. However, despite it being a promising treatment option for CTEPH, the available efficacy and safety data are based on single-arm, small-sample-size studies with large variations in patient characteristics (operable CTEPH and patients after PEA), disease characteristics, treatment approaches (BPA in addition to endarterectomy), and the duration of follow-up. The results should therefore be interpreted carefully, and further research is needed to establish the impact of BPA on survival, as well as in conjunction with medical therapy [17,18,19]. In this context, we aim to assess the efficacy and safety of the BPA procedure specifically in inoperable CTEPH patients without previous PEA or BPA in a low-volume single PH referral centre setting.

## 2. Materials and Methods

Study Population. The patients were evaluated at the Vilnius PH Referral Centre. The diagnosis of CTEPH was based on a detailed medical history, physical examination, chest radiography, chest computed tomography, transthoracic echocardiography, pulmonary ventilation/perfusion scintigraphy, right heart catheterisation (RHC), and angiographic demonstration of multiple stenoses and obstruction of the bilateral pulmonary arteries. CTEPH was diagnosed after at least 3 months of effective anticoagulation with a (a) mean pulmonary artery pressure (PAP) ≥ 25 mmHg, (b) pulmonary artery wedge pressure ≤ 15 mmHg, and (c) elevated pulmonary vascular resistance (PVR) > 3 Wood units, confirmed by right heart catheterisation according to the current guidelines [3]. The analysed cohort consisted of 26 consecutive patients who underwent BPA interventions at Vilnius University Hospital Santaros Clinics (VUHSCs) between November 2019 and December 2023. All 26 patients were receiving oral anticoagulant therapy, and 23 (88.5%) were receiving treatment with Riociguat or other targeted PH therapy. The final cohort included 12 patients, with clinical and haemodynamic data at follow-up used to evaluate the safety and efficacy of BPA in the setting of a single, low-volume centre.

Ethics Statements: This observational study was approved by the Vilnius Regional Biomedical Research Ethics Committee (approval no. 2020/1-1182-669 with amendments RL-PC-1). This research was conducted ethically in accordance with the World Medical Association Declaration of Helsinki. Only patients with inoperable CTEPH who signed informed consent to participate in the study were enrolled.

Clinical evaluation. Periprocedural tests included routine clinical practice examinations: venous blood samples measuring brain natriuretic peptide (BNP) levels; 6MWD; and transthoracic echocardiographic and RHC parameters. A 6MWD test was performed according to the ATS guidelines [20]. Two-dimensional and Doppler echocardiography measurements were conducted. The echocardiography measurements included estimated systolic pulmonary arterial pressure (sPAP); tricuspid annular plane systolic excursion (TAPSE); tricuspid annular peak systolic velocity (TAPSV); and the size and collapsibility of the inferior vena cava (IVC). sPAP was estimated by the maximal velocity of tricuspid regurgitation using the Bernoulli formula. Right atrial pressure was evaluated by assessing the IVC: it was estimated to be 3 mmHg if the vena cava was <21 mm in width and collapsed during inspiration >50%, and 8 mmHg if the vena cava was >21 mm in width and collapsed during inspiration. 

RHC was performed according to the current guidelines to invasively measure systolic (sPAP), diastolic (dPAP), and mean pulmonary arterial pressure (mPAP) and cardiac output (CO). In addition, pulmonary vascular resistance (PVR) in Wood units and the cardiac index (CI) were calculated.

BPA procedure. The procedures were performed by two specially trained interventional cardiologists at the centre with more than ten years’ experience as interventional cardiologists: one in coronary artery disease and the other in structural heart disease. 

Ultrasound-guided cannulation of either the right or left femoral vein was used to reduce puncture site complications. Sequential contrast angiography of the right and left pulmonary arteries (PAs) was performed using either a 5 Fr or 6 Fr pigtail catheter to identify stenoses. The pigtail catheter was then exchanged for a guide catheter—either an RJ, LJ, or multipurpose (MP) catheter, with the choice of 6 Fr or 7 Fr at the operator’s discretion. The decision on which PA branches to target for intervention was based on an analysis of the morphology of the PA branches and the location of the stenoses. The guiding catheter was selectively navigated into the involved branch, and the stenosis was crossed with a 0.014″ guidewire, with the use of a non-polymer “workhorse” wire recommended.

The determination of the appropriate BPA balloon was predicated on the characteristics of the vascular lesion encountered. In instances where the percutaneous coronary intervention (PCI) guidewire successfully navigated the occlusive pathology, initial dilation at the site of occlusion was undertaken with the employment of a balloon of the minimal diameter, typically 1.5 mm. Following the restoration of distal blood flow, the diameter of the vessel distally to the occlusion was evaluated, alongside an assessment for the presence of extravasation. Decisions regarding subsequent BPA interventions at the site—whether to utilise an identical balloon or opt for a marginally larger diameter—were informed by both the distal vessel diameter and the intensity of distal blood flow.

For subsequent BPA procedures, balloon selection was guided by the distal vessel diameter, often favouring a device slightly larger than the one previously employed to alleviate the occlusion. The selection of balloon diameter for stenotic sites was contingent upon the diameter of the vessel beyond the stenosis and the magnitude of the stenosis itself. For stenoses reducing the vessel’s diameter by 98–90%, a small-diameter balloon, usually 1.5 mm, was selected for the initial BPA. After dilatation, the stenosis was re-evaluated for any dissection, and it was common practice to repeat the BPA with a marginally larger balloon within the same procedural context. With each additional BPA, there was a progressive increase in balloon diameter.

Should the stenosis narrow the vessel radius by 90–75%, the diameter of the inaugural balloon was determined relative to the vessel’s diameter distal to the stenosis—typically half the vessel’s diameter. In subsequent BPAs, balloon diameter was incrementally increased to approximate the presumed distal vessel diameter. For stenoses narrowing the radius by 50–75% or less, the initial balloon’s diameter could exceed 50% of the distal vessel diameter, with subsequent balloon selections adapted to vessel size; larger vessels warranted larger initial balloon diameters, with gradual increases in subsequent sessions based on the distal vessel diameter and stenosis assessment post-dilatation. The rate of diameter increase during follow-up sessions was inversely proportional to the size of the stenosis.

Intravascular ultrasound (IVUS) was not utilised in these procedures. Stenosis and vessel diameter assessments were conducted visually or through quantitative coronary angiography (QCA), acknowledging that anatomical peculiarities, tortuosity, and suboptimal vessel support could complicate IVUS insertion. A stenosis reduction of less than 50% of the vessel radius was often deemed sufficient for optimising distal blood flow.

The balloon catheter, once inflated at the site of the stenosis, was maintained for a duration ranging from 30 to 60 s.

To prevent reperfusion pulmonary oedema, severe lesions were treated sequentially in stages over multiple BPA sessions at a mean interval of 9.5 weeks. Typically, 4–6 stenoses were treated per session, with the specific lesions selected by the interventionalist based on their complexity. The stenoses were dilated by changing the balloon diameter: the diameter of the balloon was gradually increased to the nominal diameter of the vessel in each subsequent session. As recommended by previous studies, a radiation dose of 1000 mGy, an exposure time of 60 min, and a contrast volume of 300 mL were not exceeded in a single session to minimise the risk of procedural complications. 

A 12-h post-procedure observation period was recommended, followed by discharge home if no complications were observed after 24 h. This approach ensured patient safety while making efficient use of hospital resources. 

On average, a patient underwent 6.1 BPA procedures before discontinuing treatment. The reasons for stopping BPA were improvement in mPAP; symptomatic improvement; technical difficulties; and a lack of patient compliance.

Statistical analysis. Data were analysed using repeated measures ANOVAs for 6MWD, BNP, sPAP, mPAP, CO, and PVR. Analyses were performed using JASP software (version 0.18.2) (JASP Team, 2024; https://jasp-stats.org/ accessed on 20 January 2024). The within-subject factor was the time of measurement: before the first treatment (baseline), after all treatments, and at follow-up. Bonferroni correction was used as a post hoc test to identify the specific differences between the measurement times. Sphericity assumptions were checked before the main analysis. Mauchly’s sphericity test was used to check for violations. The test yielded a significant effect that signified the violation of the assumption of sphericity; hence, Greenhouse–Geisser correction was applied. All of the following analyses were interpreted using Greenhouse–Geisser correction. Alongside the main analyses, the percentage differences between the sample means at baseline and after treatment, as well as between baseline and follow-up, were calculated for all of the variables. A *p*-value of <0.05 was considered statistically significant.

## 3. Results

Baseline characteristics of the study population. Twenty-six patients were included in this study. The mean age of the patients was 61.68 years (range 40–80), and 16 (61.5%) were male. All patients were receiving anticoagulant therapy, and 23 (88.5%) were receiving targeted PH therapy: 73.1% of these were treated with Riociguat, 7.7% with a combination of Riociguat and Bosentan, 3.8% with a combination of Sildenafil and Ambrisentan, and 3.8% with Sildenafil monotherapy. Two patients received targeted PH treatment after a single BPA procedure and were not scheduled for further procedures due to lesions in very small vessels. It is important to note that all 12 patients for whom treatment was completed and follow-up results are presented were treated with targeted PH therapy. The main characteristics of the study population are shown in Table 1. 

BPA Efficacy assessment. A total of 100 procedures were performed during the study period. Seven patients underwent only one procedure. No further procedures were planned for various reasons: mostly because the lesions were in very small vessels or at their distal end where BPA is technically impossible; the vessels were very tortuous and kinked, and there was no stenosis at maximum inspiration; or there were occlusions of the small vessels where BPA could cause significant bleeding complications.

Of the 26 patients, 14 patients completed the full course of treatment, and 12 of these have had long-term follow-up since their last BPA procedure. For these 12 patients, the mean duration of treatment from the first to the last procedure was 355.5 days (119–624). The mean interval between treatments was 67 days (28–364) (9.5 weeks). The mean follow-up from the last treatment was 258.4 days (129–364 days) or 8.6 months. The total follow-up time from the first treatment to the last treatment was 613.9 days (324–867) or 1.7 years. The mean number of sessions for the patients who completed treatment was 6.08 (range, 3–9). This longer-than-planned treatment duration may have been due to the temporary suspension of treatments during the COVID-19 pandemic.

BPA treatment is still ongoing in five patients, with two or three sessions completed so far.

In the 12 patients with a mean follow-up of 8.6 months, both 6MWD and BNP improved (a full outline of parameter dynamics per patient is presented in Table 2). The 6MWD increased from a mean of 292.9 to 394.5 metres (*p* = 0.015) after the last BPA and to 379.6 metres at follow-up (*p* = 0.043, compared to baseline), and BNP levels decreased from a mean of 590 to 111 pg/mL (*p* < 0.001) after the last BPA and to 121 pg/mL at follow-up (*p* < 0.001). In terms of haemodynamic parameters, there were statistically significant improvements in sPAP, mPAP, and PVR. mPAP decreased from a mean of 56 ± 10 to 39 ± 10 mmHg (*p* < 0.001) after the last BPA and to 37.8 ± 9.6 at follow-up (*p* < 0.001). PVR decreased from a mean of 9.8 ± 4 to 5.6 ± 2.6 Wood units (*p* < 0.001) after the last BPA and to 5.8 ± 2.9 at follow-up (*p* = 0.001) (Figure 1). There was a significant 79.5% reduction in BNP levels and a 29.7% increase in 6MWD (Figure 2). sPAP decreased by 34.4%, mPAP by 32.1%, and PVR by 40.8% at follow-up (Figure 3). The efficacy of BPA in improving haemodynamic parameters, functional capacity, and BNP levels is shown in Table 3. 

BPA Safety assessment. A total of 17 procedure-related complications occurred in 100 procedures (17% of all procedures). There were no serious or fatal complications. The most common complication was vessel dissection (10%), followed by pulmonary vascular injury and contrast extravasation and leakage into the bronchi, followed by haemoptysis, which occurred in three procedures (3%). All extravasations occurred after damaging a small-diameter vessel in an attempt to open chronic occlusions. In all cases, balloon support proximal to the site of the lesion for up to 5 min was sufficient to stop the bleeding. One case of hyperperfused pulmonary oedema was observed after opening a chronic occlusion. The patient suddenly began coughing and became short of breath as blood flow was restored distal to the occlusion. After confirming that there was no bleeding into the lung tissue, the patient did not spit out blood but continued to suffocate and become restless. The patient was immediately intubated in the X-ray operating room, and positive pressure ventilation was initiated. The procedure was halted, and the patient was transferred to the intensive care unit. The right lung was cleared of fluid (the occlusion of the upper branch of the right PA was opened during BPA). After appropriate treatment for pulmonary oedema, the patient was successfully extubated a few hours later. During the second BPA session, the remaining stenosis was re-expanded with a small-diameter balloon without complications. The complication rates decreased as the number of procedures performed increased. The complications and main procedural characteristics are listed in Table 4. 

## 4. Discussion

In this pilot low-volume single-centre study, we aimed to assess the efficacy and safety of the BPA procedure specifically in inoperable CTEPH patients. The results of the study demonstrate that the BPA procedure is an effective and safe treatment for individuals with inoperable CTEPH in a low-volume centre located in a country with a population of less than 3 million. The BPA procedure significantly improved haemodynamic parameters such as mPAP and PVR, increased exercise capacity as assessed by using 6MWD, and was associated with a significant decrease in BNP levels at the average 8.6-month follow-up. Additionally, the BPA procedures were generally safe in the low-volume centre settings, with only 17% of procedures having non-severe and non-fatal procedure-related complications, the most common being vascular dissection (10% of all sessions).

The potential novelty of this study is its insights from a low-volume centre, as well as from a very homogeneous patient cohort; i.e., all patients with inoperable CTEPH had not undergone previous attempts to perform PEA. The majority of patients were receiving targeted therapy; nonetheless, this did not mitigate the impact of the BPA. Additionally, although the follow-up period in this investigation was relatively short, it is important to note the existing gap in long-term data within this area of research. 

Efficacy of BPA. Our BPA results are comparable with the BPA results from other available studies (a summary of studies investigating the efficacy of BPA in patients with inoperable CTEPH is presented in Table A1, Appendix A). Although we analysed a small cohort of 12 patients, our cohort of inoperable patients was without any history of previous pulmonary intervention, either BPA or PEA. In most of the other analysed studies [21,22,23,24,25,26,27], between 4.5% and 42.4% of patients had undergone previous PEA. Additionally, in these studies [21,22,23,24,25], a proportion of the patients (11–13.6%) were operable but had refused to undergo PEA. The average number of sessions per patient in our study (6.1) is slightly greater than that in other studies, in which this number was between 3 and 5.2 [21,22,23,24,25,26]. The average improvement in pulmonary haemodynamics in our study is less pronounced than in the Ogawa et al. registry study [21], with a 41% and 58% decrease in PVR at follow-up, respectively; however, our cohort had a higher baseline mPAP of 56 ± 10 vs. 43.2 ± 11 mmHg in the Ogawa et al. study. The improvement in PVR is more pronounced in our study than in the study by Olsson et al. (41% and 26%, respectively) [24]. The improvement in exercise capacity is more significant in our study than in the study by Brenot et al. (30%, with a mean change of +87 metres, and 12%, with a mean change of +51 metres, respectively) [22]. It is notable that the mean baseline 6MWD is lower in our study. The reduction in BNP levels is comparable to the results of Ogawa et al. (an 80% and 82% reduction, respectively) [21].

In a recent systematic review and meta-analysis of BPA efficacy, all functional and haemodynamic parameters improved significantly following BPA in CTEPH patients [19], and these improvements are consistent with our low-volume study results. By comparison, 6MWD increased by 70 metres in the meta-analysis performed by Kennedy et. [19] as compared with 87 metres in our study; mPAP decreased by 13.2 mmHg and 18.2 mmHg, respectively; and PVR decreased by 320 dyne/cm/s^−5^ and 264 dyne/cm/s^−5^, respectively. In our study, these changes were statistically significant and are comparable with the changes seen in other studies [19,21,22,23,24,25,26,27], representing the similar efficacy of BPA not only in a high-volume tertiary PH centre setting but also in a low-volume tertiary PH centre setting.

Although not all patients in our study achieved the optimal outcome (i.e., a reduction in pulmonary artery pressure to normal), subjective clinical improvement was observed in the majority of patients treated, and one woman in her 40s had her pulmonary artery pressure reduced to near normal (the changes in the haemodynamic parameters, BNP, and 6MWD of this patient are shown in Table A2, Appendix A).

Safety of BPA. BPA is an interventional procedure and is associated with serious complications. Peri-procedural complications include vascular injury due to wire perforation and lung injury [22,28,29,30]. Comparing the mean baseline 6MWD (292.9 ± 151 and 318.1 ± 122.1), BNP levels (590 ± 445 and 239.5 ± 334.2), and mPAP (56 ± 10 and 43.2 ± 11.0) between our study population and a Japanese study population [21], it is clear that our study population had more severe disease prior to treatment initiation [21]. Brenot et al. [22] showed that a higher baseline mPAP was associated with lung injury (OR 1.08, 95% CI 1.039–1.130; *p* < 0.001), as well as a higher baseline PVR and poorer exercise capacity. Lung injury is characterised by lung opacities on chest X-ray or CT scans, with or without hypoxaemia, and it may or may not be associated with haemoptysis [29]. Lang et al. [9] also reported that patients with severely compromised haemodynamics (mPAP >40 mmHg and/or PVR >7 Wood units) have an established risk of hyperperfusion pulmonary oedema. Nevertheless, the rate of complications related to lung injury was relatively low in our study, occurring in 1% of all sessions, close to a 17.8% complication rate in the Japanese registry but higher than in the French registry (9.1%) [21,22]. These differences may be due to the avoidance of complex lesion types such as tortuous lesions and chronic total occlusions in our study. The rate of vessel injury in our study (3% of all procedures) can be compared with the results of a recent systematic review and meta-analysis of 40 studies by Kennedy et al., which showed a 5% rate of vessel injury [19].

In our study, a singular case of hyperperfused pulmonary oedema occurred following the reopening of a chronic occlusion, manifesting in sudden cough and dyspnoea as blood flow was re-established distally. The absence of pulmonary haemorrhage was confirmed, yet the patient’s respiratory distress persisted, necessitating immediate intubation in the cath lab and the commencement of lung ventilation. Proximal balloon support for up to 5 min post-lesion was adequate for haemorrhage control. The procedure was aborted, and the patient was admitted to the intensive care unit. Resolution of the pulmonary oedema was achieved through medical management, allowing for successful extubation. Subsequent BPA sessions addressed the remaining stenosis with a small-diameter balloon, without further complications.

BPA vs. medical therapy. Patients with inoperable or recurrent CTEPH can be treated with Riociguat or PH-specific drugs. The ESC/ERS recommends the use of Riociguat in symptomatic patients with inoperable CTEPH or persistent/recurrent PH after PEA (class I B recommendation) [3]. Other medical therapies may also be considered, but there is less evidence on their safety and efficacy in patients with CTEPH. The results of a systematic review and meta-analysis of 23 observational studies including 1454 patients, conducted by Wang et al. [31] and comparing BPA and Riociguat in patients with inoperable CTEPH, showed a greater improvement in pulmonary haemodynamics, New York Heart Association (NYHA) functional class, and 6MWD in patients treated with BPA compared to Riociguat. PVR was reduced with BPA by a mean difference of −1.3 Wood units (95% CI −1.57 to −1.08, *p* = 0.000) vs. −0.7 Wood units (95% CI −0.79 to −0.50, *p* = 0.000) in the Riociguat group. However, the increase in CO was greater in the Riociguat group, and there was no significant difference in the cardiac index (CI) between the groups. RACE, a French multicentre randomised controlled trial, has recently published data from CTEPH patients randomised to BPA or Riociguat [32]. At week 26, the mean pulmonary artery resistance decreased to 39.9% (95% CI 36.2–44.0) of the baseline in the BPA group and 66.7% (95% CI 60.5–73.5) of the baseline in the Riociguat group. However, the rate of complications was higher with BPA. In an ancillary follow-up study, eligible patients on first-line BPA received add-on Riociguat, and patients on first-line Riociguat received add-on BPA. At week 52, a similar reduction in pulmonary vascular resistance was observed in the patients treated with first-line Riociguat or first-line BPA (ratio of geometric means of 0.91, 95% CI 0.79–1.04) [32]. A Japanese multicentre randomised controlled trial (MR BPA) compared BPA with Riociguat [33]. At 12 months, the mean pulmonary arterial pressure had improved by −16.3 (SE 1.6) mmHg in the BPA group and by −7.0 (1.5) mmHg in the Riociguat group (group difference of −9.3 mmHg, 95% CI −12.7 to −5.9, *p* < 0.0001). However, more complications were also observed in the BPA group.

Medical therapy may add value to BPA therapy while treating different pathogenetic mechanisms of the disease. PH-specific drugs target distal and inaccessible pulmonary vasculopathy and remodelling, and BPA restores distal pulmonary flow to subsegmental levels [26]. Wiedenroth et al. [34] investigated the effects of sequential treatment with Riociguat and BPA. The results showed that Riociguat improved haemodynamics and that BPA led to further improvements. The majority of patients in the study were also receiving medical therapy, which may have confounded the results. 

BPA vs. PEA. Although some patients are only eligible for PEA and others for BPA, both techniques can treat subsegmental and segmental disease [1]. However, evidence comparing the two techniques is limited. Zhang et al. [18] compared the safety and efficacy of BPA and PEA in a systematic review and meta-analysis of 54 studies and found that BPA may have a higher perioperative and 3-year survival rate, have fewer types of associated complications, and result in a greater improvement in exercise capacity compared to PEA. However, the improvement in haemodynamic parameters was more pronounced in the PEA group compared with the BPA group (at <1-month, 1–6-month, and >12-month follow-ups). There have been no RCTs comparing the two techniques, but Ravnestad et al. [35] compared the two techniques in a single-centre observational cohort study of 82 patients. Both PEA and BPA significantly reduced mPAP and PVR, with significantly lower reductions for both parameters in the PEA group. However, there was no significant difference in exercise capacity between the groups. The results of another retrospective cohort study showed that the efficacy and safety of BPA in inoperable cases were similar to those of PEA in operable cases [36]. However, a true comparison can only be made with large, multicentre, controlled clinical trials. Long-term survival outcomes after PEA (averaging 90% at 3 years) are excellent [37], whereas long-term outcomes after BPA are lacking. However, results from a retrospective single-centre study in Japan show that long-term favourable outcomes after BPA can be expected, as haemodynamic data on mPAP and PVR were maintained over the long term (>3.5 years) [38]. Post-operative PH is observed in approximately one-quarter of patients undergoing PEA [10]. While there is limited published experience with BPA in the post-PEA setting, data from a systematic review and meta-analysis of four studies suggest that improvements in clinical and haemodynamic parameters are possible with BPA in appropriately selected patients [19]. This finding is consistent with recent guidelines and the stated recommendation of BPA treatment for residual PH after PEA [3]. The future management of CTEPH may therefore consist of both strategies and specific medical therapy, combined or sequential, adapted to the characteristics of the individual patient.

### Study Limitations

There were several limitations to this study. Firstly, this was a single-centre study with a small sample size and a relatively short follow-up period, which could impact the extrapolation of the results. Long-term results on the efficacy of BPA are therefore lacking. Secondly, almost all study participants received PH-specific medication, and there was no control group, so the effect of medication on post-procedure outcomes cannot be excluded. Thirdly, complete follow-up records were not available in all cases, which may have confounded the analyses due to missing variables. The absence of a control group in this study hampers our ability to definitively attribute the observed outcomes solely to the BPA, distinct from the effects of other concurrent treatments or the natural progression of the disease. Finally, neither investigators nor patients were blinded to treatment.

## 5. Conclusions

This pilot study demonstrates that the BPA procedure is an effective and safe treatment for individuals with inoperable CTEPH without previous PEA. We observed clinically meaningful improvements in haemodynamics and exercise capacity and a low risk of major complications in the setting of a low-volume centre. Further prospective multicentre studies are needed to compare BPA and PH-targeted therapy to guide optimal treatment strategies for patients with inoperable CTEPH.

## Figures and Tables

**Figure 1 medicina-60-00461-f001:**
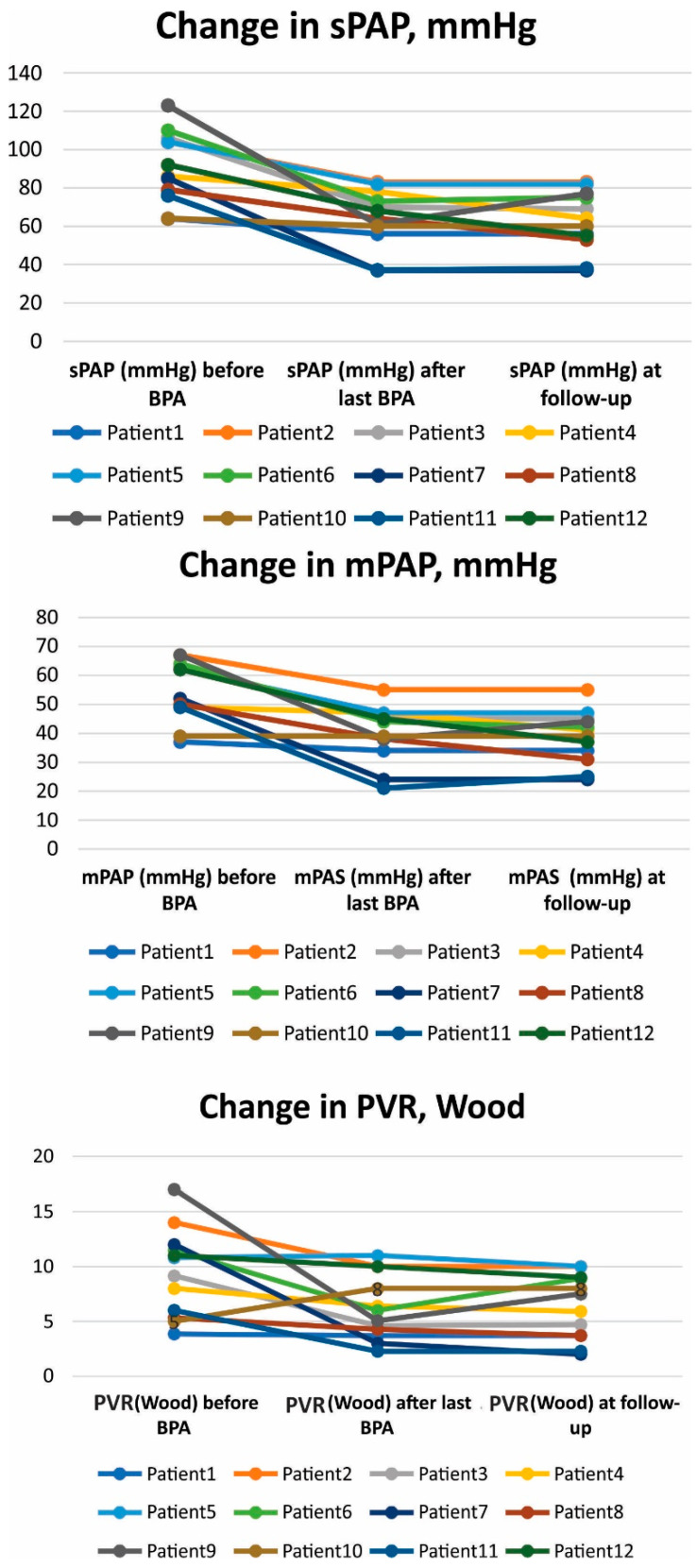
Per patient change in sPAP, mPAP, and PVR before BPA, after last BPA, and at follow-up. Abbreviations: mPAP—mean pulmonary arterial pressure; sPAP—systolic pulmonary arterial pressure; PVR—pulmonary vascular resistance.

**Figure 2 medicina-60-00461-f002:**
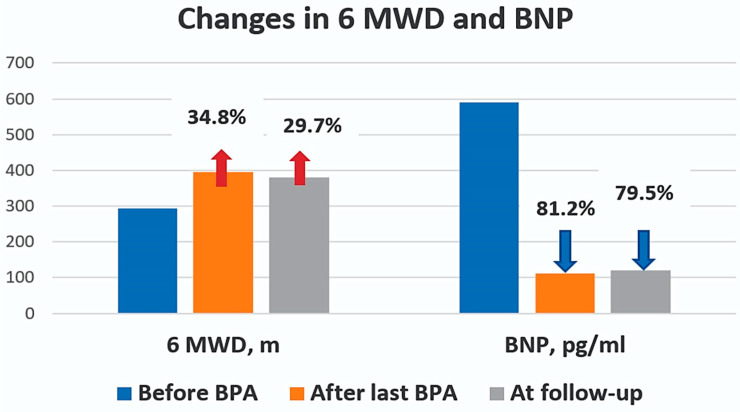
Percentage change: decrease in BNP (after the last BPA, the level of BNP decreased by 81.2% after last BPA from baseline and 79.5% at follow-up (blue arrows) and increase in 6MWD (after the last BPA, 6MWD increased by 34.8% after last BPA from baseline and 29.7% at follow-up (red arrows)). Abbreviations: BNP—brain natriuretic peptide; 6 MWD—6-min walk distance.

**Figure 3 medicina-60-00461-f003:**
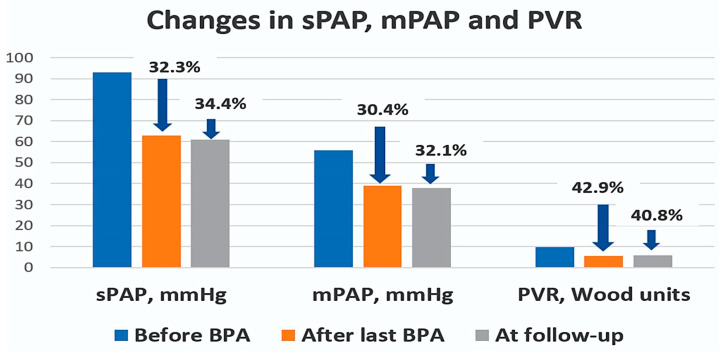
Percentage reduction in sPAP, mPAP, and PVR from pre-BPA baseline to last BPA procedure and from pre-BPA baseline to follow-up (blue arrows). Abbreviations: mPAP—mean pulmonary arterial pressure; sPAP—systolic pulmonary arterial pressure; PVR—pulmonary vascular resistance.

**Table 1 medicina-60-00461-t001:** Baseline characteristics of the study population.

Study Population, *n*	26
Mean age, years	61.68 (40–80)
Male, *n* (%)	16 (61.5)
Previous DVT, *n* (%)	12 (46.2)
History of PE, *n* (%)	21 (80.8)
Recurrent PE, *n* (%)	3 (11.5)
Current smoker, *n* (%)	3 (11.5)
Comorbidities, *n* (%)	26 (100)
➢ Hypertension	22 (84.6)
➢ Dyslipidaemia	14 (53.8)
➢ Diabetes	3 (11.5)
➢ Obesity	3 (11.5)
➢ Cancer	1 (3.8)
➢ Coronary artery disease	4 (15.4)
➢ Chronic kidney disease	2 (7.7)
➢ COPD	3 (11.5)
Targeted PH medical treatment, *n* (%)	23 (88.5)
➢ Riociguat	19 (73.1)
➢ Riociguat and Bosentan	2 (7.7)
➢ Sildenafil and Ambrisentan	1 (3.8)
➢ Sildenafil	1 (3.8)
➢ One medication	21 (80.8)
➢ Two medications	2 (7.7)
Oral anticoagulants, *n* (%)	26 (100)
➢ Warfarin	4 (15.4)
➢ Rivaroxaban	12 (46.2)
➢ Apixaban	6 (23.1)
➢ Edoxaban	4 (15.4)

Abbreviations: COPD—chronic obstructive pulmonary disease; DVT—deep vein thrombosis; PE—pulmonary embolism.

**Table 2 medicina-60-00461-t002:** Detailed parameter dynamics in patients with completed follow-up.

No/Gender	Age	Specific Treatment	Times of BPA	6 MWD (m)			BNP			sPAP (mmHg)			mPAP (mmHg)			CO (L/min)			PWR (Wood)			Mean X-ray Exposure (mGy)	Mean Procedure Time (min)
				Baseline	After BPA	At Follow-Up	Baseline	After BPA	At Follow-Up	Baseline	After BPA	At Follow-Up	Baseline	After BPA	At Follow-Up	Baseline	After BPA	At Follow-Up	Baseline	After BPA	At Follow-Up		
1/M	67	riociguat	6	510	565	600	49	21	36	64	56	56	37	34	34	6.23	5.4	5.4	3.85	3.7	3.7	679	119
2/M	59	riociguat, bosentan	6	120	180	120	975	418	387	104	83	83	67	55	55	3.8	4	4	14	10	10	809	118
3/F	54	riociguat	7	325	300	300	275	75.9	70	106	70	69	63	45	45	5.57	7.33	7	9.15	4.6	4.7	565	84
4/M	75	sildenafil	9	320	360	360	797	85	36	86	78	64	49	47	41	4.58	5.1	4.7	8	6.4	5.9	521	103
5/M	65	riociguat, bosentan	5	475	505	435	909	317	650	104	82	82	62	47	47	4.5	3.16	3.1	10.8	11	11	269	90
6/F	66	riociguat	8	300	390	376	369	160	36.1	110	73	75	64	44	42	4.27	4.5	3.57	11.47	6	8.9	545	107
7/F	40	riociguat	8	315	510	510	1035	12.1	10.5	85	37	37	52	24	24	3.06	7.11	7	12	3	2	216	105
8/M	52	riociguat	7	450	540	570	213	40	40	79	64	53	50	38	31	6,19	3.9	5.13	5.33	4.28	3.7	535	107.5
9/M	60	riociguat	6	10	520	420	1186	46.3	51.1	123	61	77	67	38	44	3.15	5,4	4,0	17	5.5	7.5	318	100
10/F	80	riociguat	4	240	240	240	161	21	26.6	64	60	60	39	39	39	4.17	2.8	3	5	8	8	323	102
11/M	70	riiociguat	4	330	340	340	22	40.5	14.4	76	37	38	49	21	25	6.3	7	7.08	6	2.28	2.28	659	105
12/F	72	riociguat	3	120	285	285	1091	97	97	92	68	55	62	45	37	4.69	3.65	2.76	11	10	9	515	101

Abbreviations: BNP—brain natriuretic peptide; CO—cardiac output; mPAP—mean pulmonary arterial pressure; sPAP—systolic pulmonary arterial pressure; PVR—pulmonary vascular resistance; 6MWD—6-min walk distance, M—male; F—female.

**Table 3 medicina-60-00461-t003:** Changes in selected clinical and haemodynamic parameters before and after treatment with BPA.

*N* = 12	Before BPA	After last BPA	*p* *	Follow-Up	*p* ^#^
6MWD, m	292.9 ± 151	394.5 ± 130	0.015	379.6 ± 138	0.043
BNP, pg/ml	590 ± 445	111.1 ± 128	<0.001	121.2 ± 195.2	<0.001
sPAP, mmHg	92.5 ± 18	62.7 ± 15.6	<0.001	60.7 ± 15.7	<0.001
mPAP, mmHg	56 ± 10	39 ± 10	<0.001	37.8 ± 9.6	<0.001
CO, L/min	4.7 ± 1.2	5.3 ± 1.3	0.749	5.0 ± 1.55	1
PVR, Wood units	9.8 ± 4	5.6 ± 2.6	<0.001	5.8 ± 2.9	0.001

Abbreviations: BNP—brain natriuretic peptide; CO—cardiac output; mPAP—mean pulmonary arterial pressure; sPAP—systolic pulmonary arterial pressure; PVR—pulmonary vascular resistance; 6 MWD—6-min walk distance; *—baseline vs. last BPA; ^#^—baseline vs. follow-up.

**Table 4 medicina-60-00461-t004:** Procedural characteristics and procedure-related complications.

Procedures, *n*	100
Range of number of procedures, mean per patient	3.84 (1–9)
Mean duration of procedure, hours	1:43 ± 0:23
Mean radiation exposure, mGy	496 ± 180
Complications, *n* (%)	17 (17)
➢ Contrast extravasation and leakage into bronchi, *n* (%)	3 (3)
➢ Haemoptysis, *n* (%)	3 (3)
➢ Vessel dissection, *n* (%)	10 (10)
➢ Hyperperfusion pulmonary oedema, *n* (%)	1 (1)

## Data Availability

Data are available in Vilnius University Hospital Santaros Clinics’ database.

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
