# Peer review of "Balloon Pulmonary Angioplasty for Inoperable Chronic Thromboembolic Pulmonary Hypertension: Insights from a Pilot Low-Volume Centre Study and a Comparative Analysis with Other Centres"

_medicina, 2024, doi:10.3390/medicina60030461_

Round 1

Reviewer 1 Report

Comments and Suggestions for Authors

I have received for review a commentary entitled “Balloon Pulmonary Angioplasty for Chronic Thromboembolic 2 Pulmonary Hypertension: Insights from a Low-Volume Centre 3 and Comparative Analysis with Other Centres” which is being processed for publication in Medicina.

I would like to congratulate the collective of authors for the proposed manuscript. The proposed case is an extremely interesting one, with therapeutic value.  Authors should pay attention to the following aspects in order to improve the proposed manuscript:

Introduction - provides an up-to-date presentation of the topic of the article that motivates the clinical study conducted

Materials and method:

- lines 71-73 - ethical issues should be systematized in a separate section

- for definitions, mention the bibliographical references according to which they were formulated

- clinical evaluation - mention in separate sections how the biological samples were evaluated, how the 6-minute walk test was performed, how the echocardiography was performed and which parameters were evaluated;

- mention the medication that was administered (including oral anticoagulants)

Results

- Table 1 - state for each medication the percentage, not just the number of patients

- figure 1 - should be enlarged to facilitate interpretation

- table A1 needs to be reorganized for ease of reading and I suggest inserting it within the results section.

Given the small number of patients I suggest to introduce the term pilot study and in the study

Discussion - perform a comprehensive review of the results of the study in comparison with those previously published in the literature.

Conclusions - needs expansion

In conclusion, the proposed manuscript brings to attention an extremely interesting topic, presenting scientific information with therapeutic value. The quality of the manuscript will be improved if the authors take into account the remarks made above.

Author Response

Dear all,

Thank you for your valuable insights. Please find answers of the authors to the reviewers’ comments below:

I have received for review a commentary entitled “Balloon Pulmonary Angioplasty for Chronic Thromboembolic 2 Pulmonary Hypertension: Insights from a Low-Volume Centre 3 and Comparative Analysis with Other Centres” which is being processed for publication in Medicina.

I would like to congratulate the collective of authors for the proposed manuscript. The proposed case is an extremely interesting one, with therapeutic value.  Authors should pay attention to the following aspects in order to improve the proposed manuscript:

Introduction - provides an up-to-date presentation of the topic of the article that motivates the clinical study conducted

Materials and method:

- lines 71-73 - ethical issues should be systematized in a separate section

Authors: Thank you. Ethical issues are systematized in a separate section now. Please, find (page 3, lines 95-99): Ethics Statements: This observational study was approved by the Vilnius Regional Biomedical Research Ethics Committee (approval no. 2020/1-1182-669 with amendments RL-PC-1). This research was conducted ethically in accordance with the World Medical Association Declaration of Helsinki. Only patients with inoperable CTEPH who signed an informed consent to participate in the study were enrolled.

- for definitions, mention the bibliographical references according to which they were formulated

Authors: Suggestion was taken into account.

- clinical evaluation - mention in separate sections how the biological samples were evaluated, how the 6-minute walk test was performed, how the echocardiography was performed and which parameters were evaluated;

Authors: We added a more detailed description of the evaluative parameters of each test now. Please, find this information in page 3, lines 100-115.

- mention the medication that was administered (including oral anticoagulants)

Authors: We have included this information now. Please, find on Page 2, lines 90-92.

Results

- Table 1 - state for each medication the percentage, not just the number of patients

Authors: We have added the percentage of medications used. Please find in Table 1, Pages 4-5.

- figure 1 - should be enlarged to facilitate interpretation

Authors: Figure 1 is enlarged now. Please find on the Page 7.

- table A1 needs to be reorganized for ease of reading and I suggest inserting it within the results section.

Authors: We reorganized Table A1 and it is in the results section as Table 2, page 6.

Given the small number of patients I suggest to introduce the term pilot study and in the study

Authors: We entitled the study as a pilot study now.

Discussion - perform a comprehensive review of the results of the study in comparison with those previously published in the literature.

Authors: This was updates according to the suggestions in the discussion section (lines 255-300).  

Conclusions - needs expansion

Conclusion section (398-403) has been corrected: This pilot study demonstrates that the BPA procedure is an effective and safe treatment for individuals with inoperable CTEPH without previous PEA. We observed clinically meaningful improvements in haemodynamics, exercise capacity, and a low risk of major complications in the setting of a low-volume centre. Further prospective multicentre studies are needed to compare BPA and PH-targeted therapy to guide optimal treatment strategies for patients with inoperable CTEPH.

Reviewer 2 Report

Comments and Suggestions for Authors

The study presents a significant contribution to the field of CTEPH, demonstrating the safety and effectiveness of BPA in a low-volume center. It highlights the potential for BPA to improve haemodynamic parameters, functional capacity, and reduce brain natriuretic peptide levels, even outside of high-volume centers.

However, several major concerns need to be addressed:

  1. -The study is limited by its single-center design and small sample size, which may affect the generalizability of the findings. Please address this point.
  2.  
  3. -The follow-up period is relatively short, raising questions about the long-term efficacy and safety of BPA. Please address this point.
  4. -The lack of a control group makes it difficult to isolate the effect of BPA from other treatments or natural disease progression. Please address this point
  5. -Almost all participants were on PH-specific medication, which could confound the effects observed post-BPA. Please address this point
  6. -The study's methodology and statistical analysis need clearer detailing to ensure the robustness of the conclusions.
  7. - I found several english errors throughout the manuscript. Please have a deep language revision.
Comments on the Quality of English Language

moderate revision required. 

Author Response

Dear all.

Thank you for your valuable insights. Please find answers of the authors to the reviewers’ comments below:

The study presents a significant contribution to the field of CTEPH, demonstrating the safety and effectiveness of BPA in a low-volume center. It highlights the potential for BPA to improve haemodynamic parameters, functional capacity, and reduce brain natriuretic peptide levels, even outside of high-volume centers.

However, several major concerns need to be addressed:

-The study is limited by its single-center design and small sample size, which may affect the generalizability of the findings. Please address this point.

-Almost all participants were on PH-specific medication, which could confound the effects observed post-BPA. Please address this point

Authors: Limitations sections has been expanded (lines 386-396).

Additionally, we have touched these points in the discussion section.

The follow-up period is relatively short, raising questions about the long-term efficacy and safety of BPA. Please address this point.

Authors: this point has been addressed (lines270-271).

-The lack of a control group makes it difficult to isolate the effect of BPA from other treatments or natural disease progression. Please address this point

Authors: this point has been addressed (lines 393-396).

-The study's methodology and statistical analysis need clearer detailing to ensure the robustness of the conclusions.

Authors: it has been corrected (please see Ethics Statement, Clinical Evaluation and Statistical analysis sections).

- I found several english errors throughout the manuscript. Please have a deep language revision. Comments on the Quality of English Language moderate revision required.

Authors: English language has been edited by the professional language editor (commercial proof-reading-service).

Reviewer 3 Report

Comments and Suggestions for Authors

The topic is interesting and the paper is quite well written. Nevertheless, in my opinion, some parts need to be improved, I have some comments:

1) Abstract. Conclusion. 28 The results of the present study demonstrated that the BPA procedure was safe and improved in- 29 vasive haemodynamic parameters, functional capacity and reduced brain natriuretic peptide levels 30 in patients with inoperable chronic thromboembolic pulmonary hypertension. The BPA procedure 31 may be effective and safe not only in high-volume, but also in low-volume centre setting. Abstract might be beneficial to include a sentence that briefly summarizes the key findings of the study. This can provide readers with a quick overview of the research. Furthermore, I suggest to improve the results section to better support the last sentence of the conclusions.

2) 1. Introduction 37 Chronic thromboembolic pulmonary disease (CTEPD) with or without pulmonary 38 hypertension (PH) is a disease caused by persistent obstruction of the pulmonary arteries 39 by organised fibrotic thrombi and associated microvasculopathy. Chronic thromboem- 40 bolic pulmonary hypertension (CTEPH) is usually considered a complication of acute pul- 41 monary embolism (PE), and it has been estimated that approximately 0.56-3% of patients 42 with acute PE develop CTEPH [1,2]. Although the Authors described in detail the findings from the included references, there are several relevant works/reviews, including most important published which should be added and discussed by the Authors:

a- Chronic Thromboembolic Pulmonary Hypertension: An Observational Study. Medicina (Kaunas). 2022;58(8):1094. doi: 10.3390/medicina58081094.

b- Balloon pulmonary angioplasty vs. pulmonary endarterectomy in patients with chronic thromboembolic pulmonary hypertension: a systematic review and meta-analysis. Heart Fail Rev. 2021;26(4):897-917. doi: 10.1007/s10741-020-10070-w.

c- Balloon Pulmonary Angioplasty for Chronic Thromboembolic Pulmonary Hypertension: A Systematic Review and Meta-analysis. Cardiovasc Intervent Radiol. 2023 Jan;46(1):5-18. doi: 10.1007/s00270-022-03323-8.

3) We aim 66 to share our first experience on BPA procedures and to evaluate the efficacy of such ther- 67 apy for inoperable CTEPH patients in a low volume PH referral centre at Vilnius Univer- 68 sity Hospital Santaros clinics. Please, improve the description of the aim of the study.

4) Table 1. Baseline characteristics of the study population. Please add the most important clinical characteristics of the study population.

5) 4. Discussion 236 In the present study, we report a series of patients with inoperable CTEPH treated 237 with BPA from November 2019. The initial results from a single tertiary centre in a small 238 country (with a population of less than 3 million) confirm that BPA is an effective treat- 239 ment for patients with inoperable CTEPH. Please, the discussion section needs to be improved.  It could be interesting to record the aim of the study. It is necessary to clarify the results obtained and compare them with previous or similar studies. 

6) 5. Conclusions 348 This pilot study represents the first experience of treating unresectable CTPH using 349 the BPA procedure in a single tertiary centre with a limited patient cohort. The BPA pro- 350 cedure was safe and effective in the setting of a low-volume centre. Further prospective 351 multicentre studies are needed to establish the efficacy of BPA in addition to specific 352 medical treatment and to assess the long-term efficacy of the procedure. Please, underline the novelty of the paper and the clinical implication of this interesting study.

Comments on the Quality of English Language

Minor changes of English language are required

Author Response

Dear all.

Please find answers of the authors to the reviewers’ comments below:

Reviewer: 1) Abstract. Conclusion. 28 The results of the present study demonstrated that the BPA procedure was safe and improved in- 29 vasive haemodynamic parameters, functional capacity and reduced brain natriuretic peptide levels 30 in patients with inoperable chronic thromboembolic pulmonary hypertension. The BPA procedure 31 may be effective and safe not only in high-volume, but also in low-volume centre setting. Abstract might be beneficial to include a sentence that briefly summarizes the key findings of the study. This can provide readers with a quick overview of the research. Furthermore, I suggest to improve the results section to better support the last sentence of the conclusions.

Authors:

Comments has been addressed (lines 22-28):

Results: Twenty-six patients with inoperable CTEPH were enrolled. The mean age of the patients was 61.6 (40–80) years. Each patient underwent a mean of 6.1 (3–9) procedures. Follow-up data were available for 12 patients. Mean pulmonary arterial pressure decreased by 32% (p<0.001) and pulmonary vascular resistance by 41% (p=0.001) at follow-up compared with baseline measurements. There was also a significant 80% (p<0.001) reduction in brain natriuretic peptide levels and a 30% (p=0.04) increase in 6-minute walk distance. The BPA procedures were generally safe in this low-volume centre setting, with only 17% of procedures having non-severe and non-fatal procedure-related complications. The most common complications included vessel dissection (10%), pulmonary vascular injury with haemoptysis (3%), and hyperperfusion pulmonary oedema (1%), which was successfully treated in all patients. Conclusions: The results of the present study demonstrate that the BPA procedure is an effective and safe treatment for individuals with inoperable CTEPH, being associated with significant improvements in hemodynamic parameters, functional capacity, and a low risk of major complications in the low-volume tertiary PH centre setting.

Reviewer: 2) 1. Introduction 37 Chronic thromboembolic pulmonary disease (CTEPD) with or without pulmonary 38 hypertension (PH) is a disease caused by persistent obstruction of the pulmonary arteries 39 by organised fibrotic thrombi and associated microvasculopathy. Chronic thromboem- 40 bolic pulmonary hypertension (CTEPH) is usually considered a complication of acute pul- 41 monary embolism (PE), and it has been estimated that approximately 0.56-3% of patients 42 with acute PE develop CTEPH [1,2]. Although the Authors described in detail the findings from the included references, there are several relevant works/reviews, including most important published which should be added and discussed by the Authors:

  • Chronic Thromboembolic Pulmonary Hypertension: An Observational Study. Medicina (Kaunas). 2022;58(8):1094. doi: 10.3390/medicina58081094.
  • Balloon pulmonary angioplasty vs. pulmonary endarterectomy in patients with chronic thromboembolic pulmonary hypertension: a systematic review and meta-analysis. Heart Fail Rev. 2021;26(4):897-917. doi: 10.1007/s10741-020-10070-w.
  • Balloon Pulmonary Angioplasty for Chronic Thromboembolic Pulmonary Hypertension: A Systematic Review and Meta-analysis. Cardiovasc Intervent Radiol. 2023 Jan;46(1):5-18. doi: 10.1007/s00270-022-03323-8.

Authors: All suggested articles have been cited in the introduction section (lines 67-75) and more comprehensively in the Discussion section (please look into the answer to 5th comment below):

Introduction: Evidence shows that BPA improves pulmonary haemodynamics, 6-minute walk distance (6MWD), functional class, and quality of life by improving pulmonary circulation [9,14–16]. However, despite it being a promising treatment option for CTEPH, the available efficacy and safety data are based on the single arm small sample size studies with large variations in patient characteristics (operable CTEPH, patients after PEA), diseases characteristics, treatments approaches (BPA in addition to endarterectomy), and duration of follow-up. The results should therefore be interpreted carefully and further research is needed to establish the impact of BPA on survival as well as in conjunction with medical therapy [17–19].

References 18 and 19 has been cited in the Discussion section before revision.

Reviewer: 3) We aim 66 to share our first experience on BPA procedures and to evaluate the efficacy of such ther- 67 apy for inoperable CTEPH patients in a low volume PH referral centre at Vilnius Univer- 68 sity Hospital Santaros clinics. Please, improve the description of the aim of the study.

Authors: Please find correction below (lines 75-77):

Introduction: In this context, we aim to assess the efficacy and safety of the BPA procedure specifically in inoperable CTEPH patients without previous PEA or BPA in a low-volume single PH referral centre settings.

Reviewer: 4) Table 1. Baseline characteristics of the study population. Please add the most important clinical characteristics of the study population.

Authors: baseline characteristics has been expanded (Table 1)

Reviewer: 5) 4. Discussion 236 In the present study, we report a series of patients with inoperable CTEPH treated 237 with BPA from November 2019. The initial results from a single tertiary centre in a small 238 country (with a population of less than 3 million) confirm that BPA is an effective treat- 239 ment for patients with inoperable CTEPH. Please, the discussion section needs to be improved.  It could be interesting to record the aim of the study.  It is necessary to clarify the results obtained and compare them with previous or similar studies.

Authors (lines 255-265): The aim of the study is included and the discussion has been improved (please find corrections below):

In this pilot low-volume single centre study, we aimed to assess the efficacy and safety of the BPA procedure specifically in inoperable CTEPH patients. The results of the study demonstrate that the BPA procedure is an effective and safe treatment for individuals with inoperable CTEPH in a low volume centre located in a country with a population of less than 3 million. The BPA procedure significantly improved haemodynamic parameters such as mPAP and PVR, increased exercise capacity as assessed by using 6MWD, and was associated with a significant decrease in BNP levels at the average 8.6 months follow-up. Additionally, the BPA procedures were generally safe in the low-volume centre settings with only 17% of procedures having non-severe and non-fatal procedure-related complications, the most common being vascular dissection (10% of all sessions).

We clarified results and compared them with other studies (please find expanded Discussion section)

Reviewer: 6) 5. Conclusions 348 This pilot study represents the first experience of treating unresectable CTPH using 349 the BPA procedure in a single tertiary centre with a limited patient cohort. The BPA pro- 350 cedure was safe and effective in the setting of a low-volume centre. Further prospective 351 multicentre studies are needed to establish the efficacy of BPA in addition to specific 352 medical treatment and to assess the long-term efficacy of the procedure. Please, underline the novelty of the paper and the clinical implication of this interesting study.

Authors: Conclusion section (lines 298-403) has been corrected: This pilot study demonstrates that the BPA procedure is an effective and safe treatment for individuals with inoperable CTEPH without previous PEA. We observed clinically meaningful improvements in haemodynamics, exercise capacity, and a low risk of major complications in the setting of a low-volume centre. Further prospective multicentre studies are needed to compare BPA and PH-targeted therapy to guide optimal treatment strategies for patients with inoperable CTEPH.

Discussion section was expanded stressing the additional value of the study (lines 266-271): The potential novelty of the study is its insights from low-volume centre, as well as a very homogeneous patient cohort, i.e. all patients with inoperable CTEPH had not undergone previous attempts to perform PEA. The majority of patients were receiving targeted therapy; nonetheless, this did not mitigate the impact of the BPA. Additionally, although the follow-up period in this investigation was relatively short, it is important to note the existing gap in long-term data within this area of research.

Reviewer: Comments on the Quality of English Language/Minor changes of English language are required

Authors: English language has been edited by the professional language editor (commercial proof-reading-service).

Round 2

Reviewer 1 Report

Comments and Suggestions for Authors

I congratulate the authors for the improved version of the proposed manuscript. Figure 1 needs to be qualitatively improved as it is not readable.

Author Response

Dear all.

Thank you for the positive review. Please find the corrected Figure 1 in the main text of manuscript.

With BR,

Authors

Reviewer 2 Report

Comments and Suggestions for Authors

Authors replied to my comments in a satisfactorily way. Ok to accept for me now.

Author Response

Dear all,

Thank you for positive review.

With BR,

Authors

Reviewer 3 Report

Comments and Suggestions for Authors

The manuscript has been improved as requested. No further comments 

Comments on the Quality of English Language

Minor changes  of English language are required

Author Response

Dear all,

The final version of the manuscript underwent professional editing service (please find certificate attached below).

Authors
